# Regulation of the Gut Microbiota and Inflammation by β-Caryophyllene Extracted from Cloves in a Dextran Sulfate Sodium-Induced Colitis Mouse Model

**DOI:** 10.3390/molecules27227782

**Published:** 2022-11-11

**Authors:** Ji Eun Yeom, Sung-Kyu Kim, So-Young Park

**Affiliations:** 1Laboratory of Pharmacognosy, College of Pharmacy, Dankook University, 119, Dandae-ro, Dongnam-gu, Cheonan-si 31116, Korea; 2SFC Bio Co., Ltd., 119, Dandae-ro, Dongnam-gu, Cheonan-si 31116, Korea

**Keywords:** β-caryophyllene, cloves, dextran sulfate sodium, inflammatory colitis, gut microbiota

## Abstract

Ulcerative colitis is an inflammatory bowel disease characterized by symptoms such as abdominal pain, diarrhea, bleeding, and weight loss. Ulcerative colitis is typically treated with anti-inflammatory drugs; however, these drugs are associated with various side effects, limiting their use. β-Caryophyllene (BCP), a natural compound derived from cloves, has antioxidant, antibacterial, and anti-inflammatory activities. In this study, we aimed to investigate the effects of BCP on colitis in a dextran sulfate sodium (DSS)-induced colitis mouse model. BCP was administered for seven days, followed by 2.5% DSS for additional seven days to induce colitis. Changes in stool weight, recovery of gut motility, colon length, colon histology, myeloperoxidase activity, inflammatory cytokines (TNF-α, IL-1β, IL-6, IgA, and IgG), and the gut microbiota were observed. Administration of BCP increased stool weight, restored gut motility, and considerably increased colon length compared to those in the untreated colitis mouse model. In addition, the amount of mucin and myeloperoxidase activity in the colon increased, whereas the concentrations of IL-1β, IL-6, and TNF-α decreased following the administration of BCP. Furthermore, BCP reduced the abundance of Proteobacteria which can cause intestinal immune imbalance. These results suggest that BCP has a potential to be developed as a preventive agent for colitis.

## 1. Introduction

Inflammatory bowel disease (IBD), a chronic inflammatory disease, is divided into Crohn’s disease and ulcerative colitis, and is characterized by symptoms such as ab-dominal pain, diarrhea, bleeding, and weight loss [1]. Anti-inflammatory drugs are used in the treatment of ulcerative colitis; however, they are associated with various side effects which limit their clinical applications [2]. Animal models of colitis have been used to determine the efficacy and toxicity of drugs in preclinical trials [3]. Mouse models of colitis induced by dextran sulfate sodium (DSS), a sulfated polysaccharide with various molecular weights, have advantages such as rapidity, simplicity, reproducibility, and controllability [4]. The DSS-induced colitis mouse model mimics human IBD with features such as diarrhea, bloody stools, and weight loss [5]. In DSS-induced colitis, the intercellular distance between crypt mucosal cells and vascular endothelial cells increases [6]. In addition, DSS affects the mucosal barrier and is toxic to intestinal epithelial cells of the basal crypt [7]. Therefore, DSS-induced colitis mouse models are indispensable for studies on imflammatory colitis.

*Syzygium aromaticum* (L.) Merr. & L. M. Perry (Syn. *Eugenia caryophyllus*) is a tree be-longing to the family Myrtaceae. Cloves, the flower buds of this tree, are widely used as spices and are commercially harvested in India, Pakistan, Indonesia, and some African countries such as Tanzania and Madagascar [8]. *S. aromaticum*-derived essential oil has been widely investigated for its antimicrobial, antioxidant, antifungal, and anti-inflammatory [9,10,11] activities. β-Caryophyllene (BCP) is a major component of many plant-derived essential oils, including *S. aromaticum*, *Betula litwinowii* (approximately 30%), and *Strobilanthus ixiocephala* (approximately 7%) [12,13,14]. BCP has antioxidant, anti-bacterial, and anti-carcinogenic [15,16,17] activities, as well as strong analgesic effects [18]. Additionally, BCP exhibits anti-inflammatory activities against gastric mucosal damage and carrageenan or prostaglandin E1-induced edema [19,20]. Cho et al. have reported that BCP attenuates DSS-induced colitis by modulating the expression of genes related to colon inflammation [21].

In this study, we explored the prophylactical potential of BCP in DSS-induced colitis by evaluating changes in the gut microbiota, stool weight, digestive tract motility, colon length, colon histology, myeloperoxidase (MPO), and inflammatory mediators.

## 2. Results

### 2.1. Evaluation of Changes in Stool Weight

Mice were randomly divided into five groups: control, 2.5% DSS, and 2.5% DSS + BCP (30, 150, or 300 mg/kg group). The body weights of the animals were measured for 9 days (Figure 1A). The DSS drinking group had significantly lower body weights than those in the control group, but oral administration of BCP did not restore the body weight loss caused by DSS. All the mice in each group survived during the experimental period until the sacrifice even though they became gaunt. In addition, for the measurement of feces, four mice from the same group were kept in one cage, feces were collected overnight, and their weight was measured. The feces did not look different, and fecal blood was not observed in each group. The fecal weight was significantly reduced in the untreated DSS drinking group compared to that in the control group. However, oral administration of BCP increased fecal weight in a dose-dependent manner compared to that in the untreated DSS-induced colitis group (Figure 1B).

### 2.2. Evaluation of Digestive Tract Motility Recovery

The contractile force of the colon segment was measured by excising the digestive tract of the animals using a physiograph. As shown in Figure 2, the contractility of the colon region was significantly reduced in the untreated DSS-induced colitis group. However, administration of BCP (150 and 300 mg/kg) restored the DSS-induced reduction in the contractile force of intestinal tissue.

### 2.3. Evaluation of Colitis Relief after Autopsy

The length of the colon was measured from the colon to the ileocecal junction from the anus, and the length from the cecum to the anus was measured. In the control group, the average colon length was approximately 6.81 cm, and when colitis was induced by 2.5% DSS drinking water, the average colon length was approximately 3.88 cm, which was significantly reduced owing to the onset of colitis. However, in the group administered with 300 mg/kg of BCP, the length of the colon was significantly recovered to an average of 5.06 cm (Figure 3).

### 2.4. Observation of Histological Changes

The observation of the colon morphology under a low-magnification microscope revealed that the four layers of the colon wall were closely attached, and the lumen structure was well maintained in the control group (Figure 4A). However, the submucosal tissue and muscle layer of the colon were widened in DSS-induced colitis mice. This effect was not altered in the BCP-treated group (Figure 4A). In addition, based on the observation of the mucous membrane in the colon tissue of the control group, the crypt structure was well-formed (Figure 4B). Crypt cells were not organized in a line, and their shape was damaged in DSS-induced colitis mice. However, this effect was reduced in BCP-treated mice.

Mucin in the colon was detected using PAS-Alcian blue staining (Figure 5A,B). Mucin was abundant in the colon tissues of the control group, whereas intestinal mucin was significantly decreased in the animals administered with DSS drinking water only (Figure 5C). Conversely, the administration of 150 and 300 mg/kg BCP significantly increased the amount of mucin in the colon (Figure 5C).

### 2.5. Observation of Immunological Changes

An average MPO activity of 1.064 U/mg was measured in the colon of the control group. In contrast, the average MPO activity was significantly increased to 5.572 U/mg after the administration of DSS (Figure 6). However, the DSS-induced increase in MPO activity was significantly decreased by the administration of 150 or 300 mg/kg BCP (Figure 6).

Additionally, the concentrations of TNF-α, IL-1β, and IL-6 in the colon tissue were detected by ELISA. BCP significantly alleviated the DSS-induced increase in the levels of TNF-α, IL-1β, and IL-6 in a dose-dependent manner (Figure 7).

### 2.6. Analysis of Changes in the Gut Microbiota

The intestinal microflora was analyzed using 16s RNA pyrosequencing, and the changes in gut bacterial composition were analyzed and compared at the phylum and family levels. As shown in Figure 8A, the relative abundance of Proteobacteria was significantly increased in the untreated DSS drinking group compared to that in the control group; however, it was reduced after the administration of 150 and 300 mg/kg BCP. In addition, the relative abundance of Enterobacteriaceae and Peptostreptococcaceae was significantly higher in the untreated DSS drinking group than that in the control group (Figure 8B). However, the administration of 300 mg/kg BCP reduced the DSS-induced abundance of Enterobacteriaceae and Peptostreptococcaceae.

The linear discriminant analysis effect size (LEfSe) method was used to compare taxa among the experimental groups to identify differentially abundant taxa. Bacilli, including Lactobacillaceae, were abundant in the control group compared to those in the untreated DSS-induced colitis group, whereas Gram-negative bacteria, including Proteobacteria, Enterobacteriaceae, and Peptostreptococcaceae, were significantly increased in DSS-induced colitis mice compared to those in control mice (Figure 9A). Moreover, Bacilli were more abundant in the 300 mg/kg BCP-administered group than in the untreated DSS-induced colitis group (Figure 9D).

## 3. Methods

### 3.1. Materials

β-Caryophyllene extracted from cloves (BCP) were provided from SFC Bio. (Cheonan, Korea), and sterile water for injection was purchased from Daihan Pharma Co., Ltd. (Seoul, Korea). DSS, colitis grade (36,000–50,000, Cat#160110) was obtained from MP Biomedicals (Santa Ana, CA, USA).

### 3.2. Ethics

The Jeonbuk National University Hospital Non-Clinical Evaluation Center was certified by AAALAC International in 2017, and this study was approved by the experimental animal ethics committee of Jeonbuk National University Hospital (IACUC No: JBUH-IACUC-2021-29).

### 3.3. Mice and Experimental Protocol

Six-week-old male C57BL/6J mice were purchased from Orientbio (Gyeonggi, Korea) and maintained in a room controlled at 20–26 °C with a relative humidity of 40–70%, a 12/12 h light/dark cycle, and 150–300 Lux. After an adaptation period of one week, the experimental animals were weighed and randomly divided into five groups with 12 animals per group. The five groups were: positive control, 2.5% DSS, and 2.5% DSS + BCP 30, 150, or 300 mg/kg group. The control mice received drinking water for 14 days. The colitis mice was pre-administered orally with drinking water for 7 days and followed by 2.5% DSS for additional 7 days. In case of test groups, the mice was pre-treated with BCP for 7 days and followed by 2.5% DSS + BCP for additional 7 days. The mice was allowed free access to 2.5% DSS. Then, 2.5% DSS was replaced by drinking water for additional 2 days. The weight change was started to measure from the 7th day of BCP administration to 16th day.

### 3.4. Observation of General Symptoms and Blood Collection by Autopsy

At the time of the autopsy, the experimental animals were anesthetized using ketamine, and cardiac blood was collected. After blood collection, the animals were euthanized, and the external surface and all orifices, cranial cavity, thoracic, and abdominal cavities, and their contents were visually inspected. The collected blood was centrifuged (1000× *g*, 10 min) to collect serum, and the serum was transferred to an Eppendorf tube and stored at −80 °C.

### 3.5. Feces Weight Measurement

After 7 days of oral administration of DSS, feces were collected once before and after the replacement of normal drinking water. The experimental animals were placed in metabolic cages and feces were collected once a day. The weight of feces was measured immediately after collection.

### 3.6. Assessment of Colon Length and Contraction Reaction

Immediately after blood collection, the colon was separated, the blood and fat on the surface were removed using physiological saline, moisture was removed using absorbent paper, and the colon length was measured. After measurement, tissues were stored at −80 °C for further use, and colons were placed in formalin solution and fixed at room temperature for histological analysis.

The colon tissue was added to a Krebs solution (18 mM NaCl, 4.8 mM KCl, 2.5 mM CaCl_2_, 1.2 mM KH_2_PO_4_, 1.5 mM MgSO_4_, 25 mM NaHCO_3_, and 11 mM glucose; pH 7.4) saturated with a gas mixture (95% O_2_ + 5% CO_2_) at room temperature. Mesenteric tissue was removed in Krebs solution and cut to a length of 1–2 cm to prepare a colon segment. The colon segment was placed in a chamber filled with Krebs solution maintained at 37 °C, and one end of the colon segment was connected to a tension transducer. The contraction force was measured and recorded using a physiograph recorder connected to force transducers and a PowerLab data-acquisition system. Equilibrium was maintained until the temperature and excitability became constant over 60 min by applying 1 g basal tension. Carbachol or phenylephrine was administered in the Krebs solution chamber. The contraction response (contraction force, contraction frequency, etc.) of the colon segments was measured.

### 3.7. Histopathological Examination

Colon tissues extracted from the mice were fixed using 10% formalin. After fixation, a paraffin block was prepared by dehydration and embedding, and a coronal segment with a thickness of 5 μm was prepared. Paraffin in the colon tissue sections was removed using xylene solution and hydrophilized with 100%, 95%, 90%, 80%, and 70% ethanol. The sections were then stained with hematoxylin and eosin and observed under an upright optical microscope (Olympus, Tokyo, Japan).

Similarly, colonic tissue sections were prepared and stained with PAS-Alcian blue. This is a method of staining acidic mucin polysaccharides with blue and neutral mucin with bright magenta, and it was performed according to the manufacturer’s instructions.

### 3.8. Immunological Evaluation

After euthanizing the experimental animals, a 1 cm area at the end of the colon was excised and washed with phosphate-buffered saline (PBS). A 10 mM PBS (pH 7.0) solution containing 0.5% hexadecyl-trimethyl-ammonium bromide (HETAB) and 10 mM ethylenediaminetetraacetic acid (EDTA), corresponding to 10 times the tissue volume, was added and homogenized using a homogenizer. After freezing in liquid nitrogen, it was thawed again at 37 °C, and this process was repeated thrice. The section was crushed using a sonicator and centrifuged at 20,000 g and 4 °C for 30 min. The supernatant was collected for measurement of enzymatic activity. The enzyme reaction solution was prepared by mixing the following mixture in a tube: 80 mM PBS (pH 5.4) containing 0.5% HETAB, 10 mM TMB (3, 3′, 5, 5′-tetramethyl benzidine), and 15% hydrogen peroxide (H_2_O_2_). A sample was added to the enzyme reaction solution and mixed using a vortex mixer, followed by an enzymatic reaction at 37 °C for 3 min. After the reaction was terminated by adding 0.2 mM sodium acetate buffer (pH 3.0), the optical density was measured at 655 nm. A standard curve was prepared using purified MPO (myeloperoxidase) with known active units as a standard, and the MPO activity of the sample was calculated by comparing it with the absorbance value of the sample reaction solution. MPO activity was expressed in units/g.

To identify inflammatory cytokines, colonic tissue was placed in a lysis buffer and homogenized, followed by centrifugation at 13,000× *g* for 10 min. Interleukin (IL)-1β, IL-6, and TNF-α were measured using the supernatant according to the ELISA kit manual. Briefly, coating solution was parepared by diluting the capture antibody in coating buffer. The plates were coated with 100 µL per well of coating solution for 1 h at room temperature or 12–18 h at 2–8 °C. Then, the paltes were washed with 300µL of wash buffer per well, and blocked with 300 µL per well with blocking buffer for 1 h at room temperature. One hundred µL of standards and samples were added into designated wells, and incubated at room temperature with gentle continual shaking. After 2 h, the plates were washed and 100 µL of the detection antibody solution was added into each well. After 2 h incubation, the working solution of streptavidin-HRP in Blocking buffer was added into each well, and incubate for additional 1 h at room temperature with gentle continual shaking. After the rinse, 100 µL of TMB substrate solution was added to each well, and the plate was incubated for 30 min at room temperature. After adding 100 µL of stop solution, the absorbance was measured at 450 nm.

### 3.9. Pyrosequencing Analysis of the Gut Microbiota Based on the 16S rRNA Gene

Stool samples were collected 14th day after administration of the test substance for gut microbiome analysis. Metagenomic DNA was extracted according to the manufacturer’s instructions using the Fast DNA SPIN Kit (MP Biomedicals). First, the collected sample was lysed in 1 mL of cell lysis buffer and centrifuged at 14,000× *g* for 5–10 min to pellet debris. Then, the supernatant was transfered to a 2.0 mL microcentrifuge tube. and an equal volume of Binding Matrix was adeed. After the incubation with gentle agitation for 5 min at room temperature, the half of the suspension was transfered to a SPIN™ Filter, and centrifuged at 14,000× *g* for 1 min, and this step was repeated one more time. The 500 µL of prepared SEWS-M was added to the pellet and it was resuspended. After the centrifugation at 14,000× *g* for 2 min by replacing the catch tubes, DNA was eluted by gently resuspending the Binding Matrix above the SPIN filter in 100 µL of DES. After incubating for 5 min at 55 °C in a heat block or water bath, the tube was centrifuged again at 14,000× *g* for 1 min to spin down eluted DNA into the clean catch tube. The collected DNA was stored at −20 °C for use. The generated metagenomic DNA samples were analyzed by measuring the absorbance at 260 nm.

### 3.10. Statistical Analysis

Statistical analysis was performed using GraphPad Prism statistical program (Ver. 8.0) for all data, including the weight of the experimental animals and the length of the tissue. An unpaired *t*-test was performed to confirm the significance between the test groups. When the unpaired t-test was performed, the Mann–Whitney test was performed when n was ≥5, and Welch’s test was performed when n ≤ 4. The confidence interval (CI) was set at 95%, and all data are expressed as mean ± SEM.

## 4. Discussion

Researches using natural materials are continuously increasing, and the new approaches to improve the biological activity of natural products are being conducted very actively for the treatment and prevention of diverse diseases [22,23]. BCP is found in spices, fruits, and medicinal essential oils, and is approved by the FDA and in Europe as a food additive, flavor enhancer, and flavoring agent [24]. In this study, we revealed that BCP attenuated DSS-induced colitis in C57BL/6J mice by regulating the gut microbiota and inducing macroscopic, histological, and immunological changes.

First, we observed the macroscopic effects of BCP on DSS-induced colitis. In the colitis model, the length of the colon decreases as the disease worsens [25]. In our study, BCP significantly recovered colon length in a dose-dependent manner and recovered fecal weight. In addition, BCP restored the contractile force of the intestinal tissue in a dose-dependent manner, suggesting that BCP improved intestinal motility. Although weight loss in animals is directly related to disease severity [26], BCP had little effect on alleviating DSS-induced weight loss. This effect might be because the colon becomes thicker as it becomes shorter [27]. These results suggest that BCP exhibits a significant preventive effect on ulcerative colitis.

The colon wall is composed of the mucous membrane, intestinal mucosal epithelium, submucosal tissue, muscle, and serous matrix. In the DSS-induced ulcerative colitis model, colonic epithelial cell death occurs, and the mucosal layer is lost, leading to inflammatory response [28]. In our study, the gap between the submucosal tissue and the muscle layer in the DSS-induced colitis model was not restored by the administration of BCP.

Mucin, a major component of the mucus membrane surrounding the gastrointestinal tract, is a high-molecular-weight glycoprotein that is glycosylated with O-linked oligosaccharides and N-glycan chains linked to the protein backbone [29]. The secreted mucin forms a protective boundary between a loosely attached mucus layer that provides a niche for enterobacteria and a tightly attached mucin layer that is free of bacteria [30]. Intestinal microbes play an important role in maintaining intestinal homeostasis by establishing and maintaining beneficial interactions with mucosal immune cells and intestinal epithelial cells. Damaged or dysfunctional mucosal barriers promote immune responses [31]. In our study, the administration of BCP significantly increased the amount of mucin in the colon in a dose-dependent manner, indicating that it is effective in alleviating ulcerative colitis from a histological point of view.

MPO is an enzyme mainly expressed in neutrophils, and is activated when an acute inflammatory response is induced [32]. The acute inflammatory response that occurs in mice with DDS-induced colitis is at least partially related to the overexpression of the inflammatory cytokines IL-1β, IL-6, and TNF-α [33,34]. In this study, BCP significantly decreased the MPO activity in a dose-dependent manner. Furthermore, the levels of inflammatory mediators and cytokines, including TNF-α, IL-1β, and IL-6, were significantly lower in the BCP group than those in the untreated DSS-induced colitis group. These results suggest that BCP has a significant effect on the immunological changes in DSS-induced ulcerative colitis.

An imbalance of the gut microbiota is associated with the onset and pathogenesis of ulcerative colitis [35]. Proteobacteria with adherent and invasive properties are complexly involved in luminal dysbiotic changes, exploit host defenses, and induce proinflammatory changes to favor dysbiosis in the gut microbiota [36]. In addition, Peptostreptococcaceae and Enterobacteriaceae are abundant in the ulcerative colitis model, and this abundance is related to an increase in the inflammatory response [37,38]. In this study, oral administration of BCP reduced the abundance of Proteobacteria, Enterobacteriaceae, and Peptostreptococcaceae. These results suggest that BCP alleviates DSS-induced ulcerative colitis by regulating the gut microbiota.

In conclusion, our results revealed that oral administration of BCP in DSS-induced ulcerative colitis mice improved colitis by reducing MPO activity, IL-1β, IL-6, and TNF-α concentrations, as well as the abundance of bacteria that cause intestinal immune imbalance. These data suggest that BCP can be used as a preventive agent for ulcerative colitis.

## Figures and Tables

**Figure 1 molecules-27-07782-f001:**
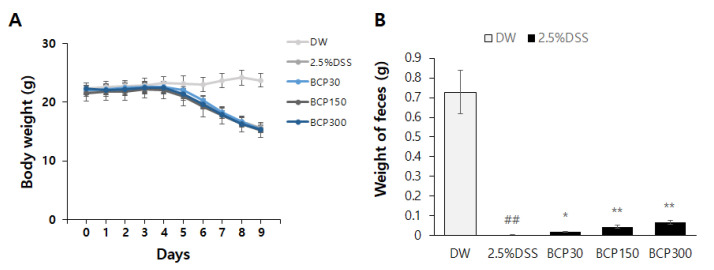
The body weight and feces weight following the administration of β-caryophyllene (BCP) in the dextran sulfate sodium (DSS)-induced colitis model. The body weight (**A**) and feces weight (**B**) were measured and evaluated in each group. Data are presented as mean of ± SEM. Group comparisons were accomplished using one-way ANOVA test and unpaired *t*-test with post-test (##, *p* < 0.01, compared to vehicle-treated control mouse) (*, *p* < 0.05; **, *p* < 0.01, compared to vehicle-treated, 2.5% DSS-induced colitis mouse).

**Figure 2 molecules-27-07782-f002:**
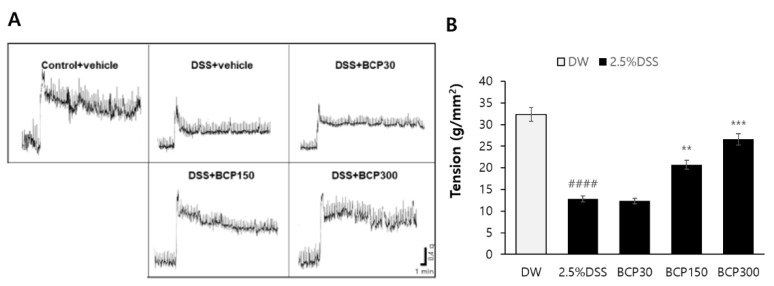
The effect of BCP on the changes of the inotropic contractile force in DSS-induced colitis model. After colon collection, the inotropic contractile force of the colon was measured to compare the effect of BCP on intestinal motility during colitis. Data are presented as mean ± SEM. Group comparisons were performed using unpaired *t*-test with post-test (####, *p* < 0.0001, compared to vehicle-treated control mouse) (**, *p* < 0.01; ***, *p* < 0.001, compared to vehicle-treated, DSS-induced colitis mouse).

**Figure 3 molecules-27-07782-f003:**
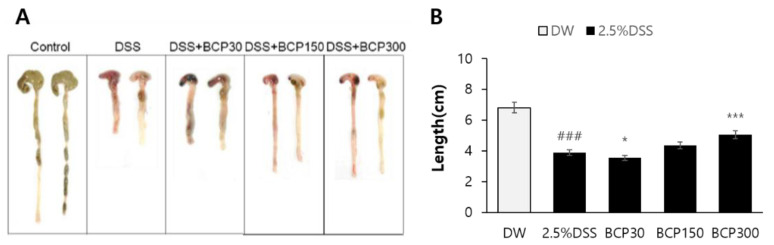
The effect of BCP on the length of the colon in a DSS-induced colitis mouse model. After collecting the colon, its length was measured from the proximal colon to the distal colon. Data are presented as mean ± SEM. Group comparisons were performed using unpaired *t*-test with post-test (###, *p* < 0.001, compared to vehicle-treated control mouse) (*, *p* < 0.05; ***, *p* < 0.001, compared to vehicle-treated, DSS-induced colitis mouse).

**Figure 4 molecules-27-07782-f004:**
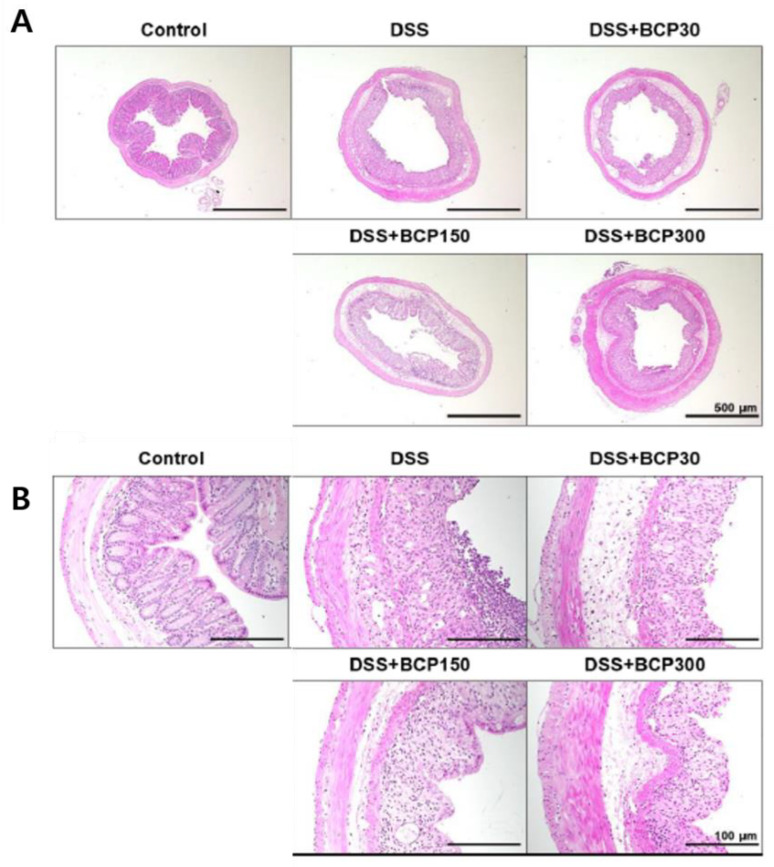
Hematoxylin and eosin staining of mouse colons for evaluating the effect of BCP administration in DSS-induced colitis. The colonic tissure sections were observed at (**A**) 40× and (**B**) 200× magnification.

**Figure 5 molecules-27-07782-f005:**
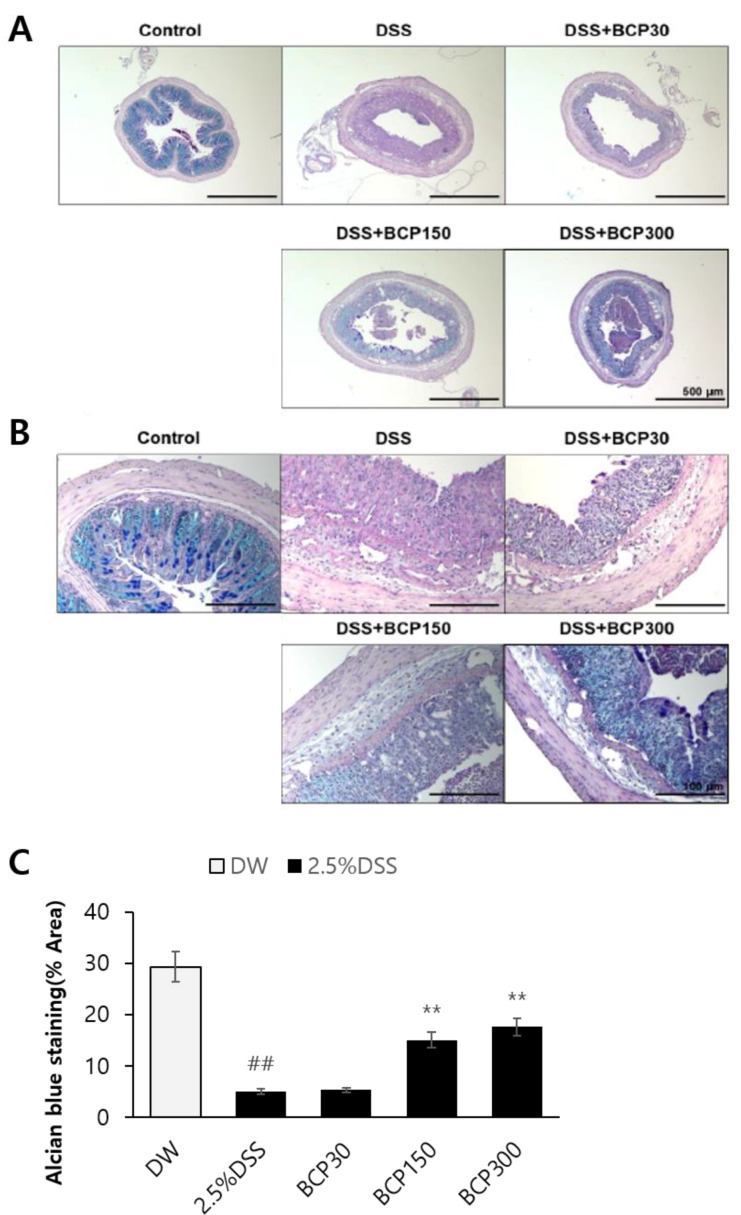
PAS-Alcian blue staining in mouse colons to evaluate changes in mucin in DSS-induced colitis model. The colonic tissure sections were observed at (**A**) 40× and (**B**) 200× magnification. PAS-Alcian blue staining was performed to compare (**C**) the contents of mucus in colons. Data are presented as mean ± SEM. Group comparisons were performed using unpaired t-test with post-test (##, *p* < 0.01, compared to vehicle-treated control mouse) (**, *p* < 0.01, compared to vehicle-treated, DSS-induced colitis mouse).

**Figure 6 molecules-27-07782-f006:**
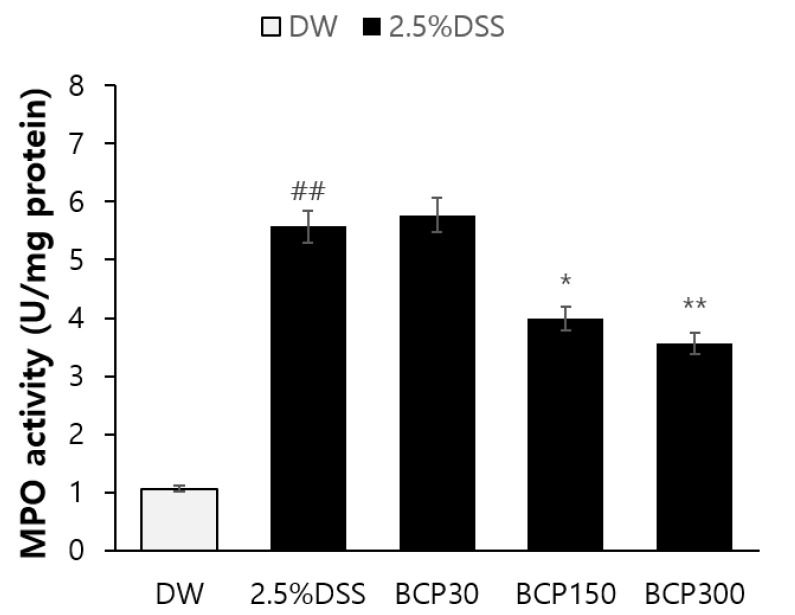
The activity of myeloperoxidase (MPO) in mouse colons. MPO activity in the colon was measured as an immunological parameter for acute inflammation. Data are presented as mean ± SEM. Group comparisons were performed using unpaired t-test with post-test (##, *p* < 0.01, compared to vehicle-treated control mouse) (*, *p* < 0.05; **, *p* < 0.01, compared to vehicle-treated, DSS-induced colitis mouse).

**Figure 7 molecules-27-07782-f007:**
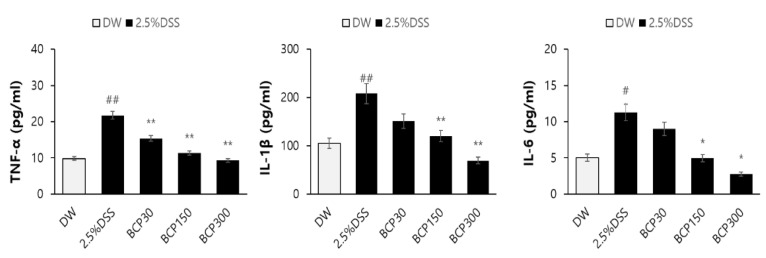
ELISA analysis of pro-inflammatory cytokines in mouse colons. The levels of the pro-inflammatory cytokines, TNF-α, IL-1β, and IL-6 were measured. Data are presented as mean ± SEM. Group comparisons were performed using unpaired t-test with post-test (#, *p* < 0.05; ##, *p* < 0.01, compared to vehicle-treated control mouse) (*, *p* < 0.05; **, *p* < 0.01, compared to vehicle-treated, DSS-induced colitis mouse).

**Figure 8 molecules-27-07782-f008:**
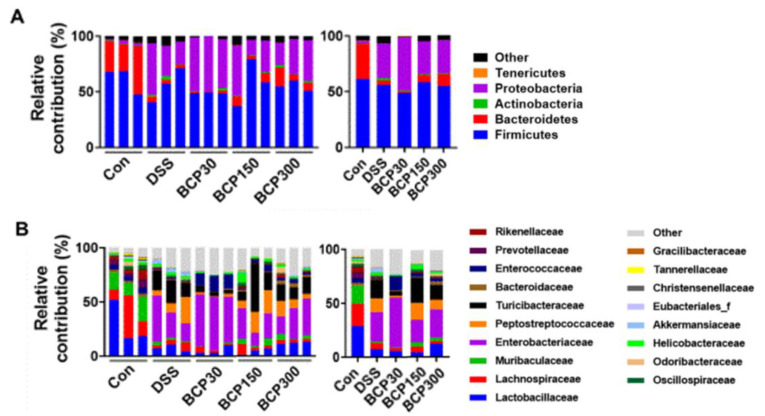
Taxonomic composition (percentage) of the gut microbiota of the experimental groups at the Phylum and Family levels. The gut microbiota were classified as (**A**) Phylum, and (**B**) Family.

**Figure 9 molecules-27-07782-f009:**
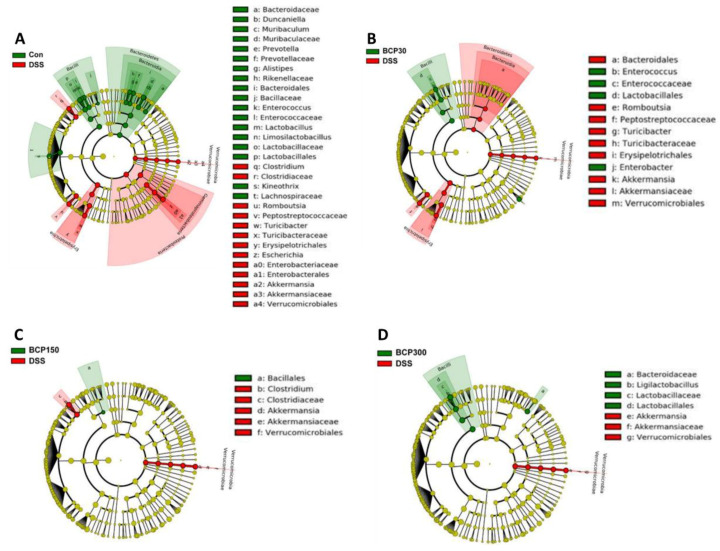
Linear discriminant analysis effect size of the microbiome in mouse feces of (**A**) untreated control group vs DSS-treated group and (**B**) 30, (**C**) 150 and (**D**) 300 mg/kg BCP + DSS-treated group vs DSS-treated group.

## Data Availability

MDPI Research Data Policies at https://www.mdpi.com/ethics.

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
