# Peer review of "Regulation of the Gut Microbiota and Inflammation by β-Caryophyllene Extracted from Cloves in a Dextran Sulfate Sodium-Induced Colitis Mouse Model"

_molecules, 2022, doi:10.3390/molecules27227782_

Round 1

Reviewer 1 Report

The work entitled ''Regulation of the gut microbiota and inflammation by β-caryophyllene extracted from cloves in a dextran sulfate sodium-induced colitis mouse model'' by Yeom et al. seems to be a solid study of the biological activity of a natural product derived from the natural extract. Mouse model has been used for testing.   Just to make this study more interesting to a broader spectrum of readers, authors should refer to some new analogical studies that were recently published. Papers that could help them in it are given below:   Molecules 27 (18), 5744   and this one:   Pharmaceutics 14 (8), 1711   Moreover, the quality of the Fig. 8 should be improved. In its current form it is barely readable.   Authors should elaborate a little bit more on the way that was used to calculate the error bars presented on the diagram within the paper.   These changes will improve the paper significantly.

Reviewer 2 Report

Dear editor; The attached articled was checked. The manuscript contain interesting information about  Regulation of the gut microbiota and inflammation by β-caryophyllene extracted from cloves in a dextran sulfate sodium-induced colitis mouse model

I think that this article well suits to your journal.

It is generally a good work. The scientific and presentation level of manuscript is high. 

Methodology is intelligible

References were cross-checked.

-The paper should be edited according to the writing rules of the journal.

Reviewer 3 Report

The manuscript describes the effects of beta-caryophyllene on a dextran sulphate sodium induced ulcerative colitis mouse model. Histopathological parameters, levels of cytokines, analysis of gut microbiota have been evaluated.

Results indicate that caryophyllene reduces inflammation in the investigated model.

Major Weaknesses are in the description of methodology:

The authors declare to test beta-caryophyllene but in Methods they report 'cloves extract'. How do they isolate beta-caryophyllene from cloves extract? Or they administer the extract? It is very important to clarify this aspect.

More details on the methodology used to quantify cytokines must be afforded.

More details on the analysis of gut microbiota must be afforded.

In 'Results' paragraphs 3.3 and 3.4 have the same title. Titles must be differentiated and referred to the content of the paragraphs.
